# Ethical and Forensic Issues in the Medico-Legal and Psychological Assessment of Women Asylum Seekers

**DOI:** 10.3390/healthcare11172381

**Published:** 2023-08-24

**Authors:** Valeria Tullio, Corinne La Spina, Daniela Guadagnino, Giuseppe Davide Albano, Stefania Zerbo, Antonina Argo

**Affiliations:** 1Section of Legal Medicine, Department of Health Promotion, Mother and Child Care, Internal Medicine and Medical Specialties, University of Palermo, 90129 Palermo, Italy; corinne.laspina@unipa.it (C.L.S.); daniela.guadagnino@you.unipa.it (D.G.); giuseppedavide.albano@unipa.it (G.D.A.); stefania.zerbo@unipa.it (S.Z.); antonella.argo@unipa.it (A.A.); 2Interdepartmental Center of Research (CIR) on Migration, University of Palermo, 90129 Palermo, Italy

**Keywords:** women, asylum seekers, violence and torture, legal medicine, forensic, clinical psychology

## Abstract

Asylum-seeking migrants represent a vulnerable segment of the population, and among them, women constitute an even more vulnerable group. Most of these women and girls have been exposed to threats, coercion, and violence of many kinds, including rape, forced prostitution, harassment, sexual slavery, forced marriage and pregnancy, female genital mutilation/excision, and/or other violations of their rights (e.g., deprivation of education, prohibition to work, etc.). The perpetrators of the violence from which they flee are often their own families, partners, and even institutional figures who should be in charge of their protection (such as police officers). In the process for the acceptance/rejection of an asylum application, the forensic and psychological certification can make the difference between successful and unsuccessful applications, as it can support the credibility of the asylum seeker through an assessment of the degree of compatibility between the story told and the diagnostic and forensic evidence. This is why constant and renewed reflection on the ethical, forensic, and methodological issues surrounding medico-legal and psychological certification is essential. This article aims to propose some reflections on these issues, starting from the experience of the inward healthcare service dedicated to Migrant Victims of Maltreatment, Torture, and Female Genital Mutilation operating since 2018 at the Institute of Forensic Medicine of the University Hospital of Palermo.

## 1. Introduction

The number of refugees worldwide is growing dramatically. A trend that is expected to continue due to a global increase in social, economic, and political instability, conflicts, and the consequences of climate change [1,2,3,4]. In recent years, the female migrant population towards EU countries has steadily increased, and the so-called “phenomenon of feminisation of migration flows” has long been the focus of numerous studies [5]. As far as Italy is concerned, statistics show that from 2020 to 2021, the number of women asylum seekers has grown by 65% [6,7], with a predictable upward trend given the number of migrants who have landed on the coasts of southern Italy, and Sicily in particular, in the last year [8]. These women and girls (sometimes minors) constitute a particularly vulnerable segment of the migrant population. They are often women who seek asylum because they are fleeing wars, arranged marriages, or family violence and have undergone or fear undergoing Female Genital Mutilation (FGM) [9,10,11,12]; they are victims of sexual violence (in the country of origin, during the migration route, during the period of imprisonment/slavery in Libya) [13,14]. In addition, in most cases, during the migration route, they become victims of trafficking and are forced into prostitution. It is crucial for asylum seekers to be able to provide evidence of persecution/torture/violence suffered in their countries of origin or during the migration route, and the case-by-case assessment of thousands of asylum applications is one of the most significant challenges for host countries [1]. In this context, the medical-legal and psychological certifications, increasingly requested by Asylum Application Assessment Commissions and by lawyers assisting migrants, are particularly relevant. These certifications, by helping to support or not support applicants’ accounts from a diagnostic and forensic perspective, can make the difference in the acceptance or rejection of applications [15,16,17], and therefore constant attention to the ethical, forensic, and methodological issues inherent in the assessment process appears indispensable. Starting from the experience of the inward healthcare service dedicated to Migrant Victims of Maltreatment, Torture, and FGM active at the Institute of Forensic Medicine of the University Hospital of Palermo, the aim of this article is to focus on some ethical, psychological, and forensic issues relating to the process of medico-legal/psychological assessment, with particular reference to migrant women applicants for international protection who have arrived in Sicily from Africa (mainly sub-Saharan).

## 2. Our Inward Healthcare Service

The inward healthcare service dedicated to Migrants Victims of Maltreatment, Torture, and FGM started in 2018 at the Institute of Forensic Medicine of the University Hospital of Palermo. The service is mainly accessed by migrants who have applied for international protection/asylum and are therefore waiting to be interviewed by the Territorial Commissions assessing asylum applications or, if their application has been rejected, by migrants who are waiting for an appeal hearing. The medical-legal/psychological certification is mainly requested by asylum seekers’ lawyers, by the local general medicine clinic dedicated to migrants, by the referrers of Reception Centers, by voluntary associations, and by referrers of non-governmental associations active in the area (mainly Médicine Sans Frontières, MSF). In some cases, the examination is carried out directly at the request of the Territorial Commissions. The healthcare team carrying out the examination comprises a medico-legal expert, a psychologist, and a linguistic-cultural mediator; if the applicant is a minor, the legal guardian is present. At the request of the person concerned, the applicant’s lawyer may also be present. Before starting the meeting, the applicant is informed in detail about the characteristics, procedures, and reasons for the visit and signs the informed consent form. For the assessment, the reference used is the Istanbul Protocol [18,19]. The key instrument for documenting and reporting torture is a United Nations document that sets internationally recognized standards for the medical examination of persons claiming to have been subjected to torture and ill-treatment [18,19,20,21].

### Women Accessing Our Inward Healthcare Service

From 2018 to 2022, 43 women between 14 and 56 years of age, mainly from sub-Saharan Africa and Tunisia, were examined at the inward healthcare service for medical-legal certification. The women who underwent the medico-legal assessment were fleeing from conditions of slavery and domestic violence as a result of forced marriages. They flee from violence suffered by their parents because they refused an arranged marriage and from extreme poverty [22,23] (which makes the women’s condition even more unbearable than men, because most African women have greater difficulty in accessing finance or loans for starting work activities, the profound gender disparities in work pay (when present), their impossibility (in some countries) to inherit and/or maintain possession of family assets (house, livestock, farmland, etc.), and the fact that they were victims of rape and therefore considered a disgrace by their families. Some have been sold to traffickers by their families or by people they trusted; others still leave because they have been deceived by traffickers and their accomplices into believing that by leaving their country they would find a good job and better living conditions. Almost all of them have a low or no level of education: 71% of women attended primary school; 14% are illiterate; 6% attended school for between 6 and 10 years; and 9% attended school for more than 10 years. The 24% of these women have children who remained in their home country; 90% did not have a husband/partner at the time of their departure; and 10% fled because of forced marriages. The 57% of them were subjected to FGM in childhood; 72% suffered atrocious violence while in Libyan prisons or were forced into prostitution for years.

## 3. Medico-Legal/Psychological Certification in the Asylum Application Process

The importance of the medico-legal/psychological assessment of the psycho-physical outcomes of intentional violence in asylum seekers and the risks related to deficiencies in the forensic assessment is internationally recognized [15,16,18,19,20,21,22,23,24,25,26], and provides for a judgment of “compatibility” between the asylum seeker’s narrative of the events that occurred and the physical and psychic signs/outcomes/sequelae detectable at the time of the visit. 

In Italy, the medico-legal/psychological act is part of the long process aimed at accepting/rejecting asylum applications, which involves different institutional components (Territorial Commissions, Police Forces, Courts) and professional figures (forensic doctors, psychologists, cultural mediators, lawyers, health personnel, social workers, Non-Governmental Organization personnel, Community and Reception Center personnel, legal guardians in the case of minors, etc.) who intervene at different or overlapping stages of the process, sometimes with conflicting purposes. The process to be followed by asylum seekers (which differs from one country to another but whose basic procedure is the same in Europe) (Figure 1) [27] is complex, often lengthy, and, due to the way it is structured, contains within itself an ineliminable underlying paradoxical condition. 

The process exposes the subject whose condition of fragility/vulnerability is to be ascertained, for the purposes of granting protection and international protection, to the concrete risk of re-traumatization (traumatic stress reactions, emotional, and/or physical, triggered by exposure to memories or reminders of past traumatic events) due to the repeated re-evocations of the traumatic events experienced. In addition, the process exposes the subjects to re-victimization due to the repeated levels of “judgement” on his credibility, on the “real gravity” of the motivations that led them to migrate, and on the assessment of the “degree of severity” of the physical and psychic sequelae of what was suffered in the country of origin and/or during the migration journey. 

In the asylum application process, moreover, it is the applicant who must produce evidence that he or she meets all the requirements, evidence of torture/violence suffered, that there is, as the Geneva Convention (1951) [28] states, a “well-founded fear of being persecuted for reasons of race, religion, nationality, membership of a particular social group or political opinion”, and that persecution will resume if he or she returns to the country of origin. Furthermore, a link between the persecutory act and the applicant’s race, religion, nationality, political opinion, or membership in a particular social group must be demonstrable [26,29]. Often, however, the only evidence an individual may have is his or her testimony, and as pointed out by some authors [30], asylum seekers may feel under pressure not only to prove persecution and violence but also to prove that harm resulted from them. If, for some, proving that they have suffered physical violence/torture (and consequent “harm”) is difficult because they cannot testify to the marks on their body, it will be even more complicated to prove psychological violence/torture [31,32,33], psychological sequelae of what they have suffered, and the “well-founded fear of persecution” referred to in the Geneva Convention (1951) [28]. This is even though the very use of the term “fear” in the text of the Convention implies that a refugee is defined to some extent by his or her psychological response to events [34] and that the “wounds of the soul” have the same importance, severity, and pervasiveness as the physical ones. The issue of assessing the credibility of allegations of torture based on medical and psychological criteria then becomes central, to the point of being one of the key issues also addressed in the Istanbul Protocol [18,19]. 

Sometimes, the Territorial Commissions and the legal system do not adequately take into account the psychological dimension of asylum seekers, and how their trauma affects their ability to provide credible testimony. The difficulty that individuals may have in recounting a coherent narrative of highly stressful or traumatic situations, therefore, ends up backfiring on traumatized asylum seekers [25,35]. Even if an applicant’s story meets all the legal requirements of the refugee definition or his/her credibility is questioned on even a minimal issue, the claim may be rejected [26]. It is also necessary to carefully consider how much and how possible distortions in evaluations of migrants’ positions resulting from unconscious stereotypes or prejudices of the adjudicator, linguistic misunderstandings, or cultural differences may affect the acceptance or rejection of the asylum application. In this context, a correct medical-legal/psychological assessment, conducted following fundamental ethical principles and the principle of the ‘third party’ role, is a fundamental part of the legal process. Moreover, it is not uncommon that during medical/psychological assessments, applicants reveal further details and episodes of trauma in addition to those reported in the preliminary statement made at the time of the formalization of the asylum application or during the hearing at the Territorial Commission [15].

## 4. Critical Issues

The Istanbul Protocol [19] states that all health professionals, therefore also forensic doctors and psychologists, who undertake clinical and forensic evaluation in cases of alleged or suspected torture/ill-treatment must always operate according to the fundamental ethical principles of “*to act in the best interests of patients (beneficence), ‘do no harm’ (non-maleficence), respect the decisions of patients (autonomy) and maintain the confidentiality of information shared in encounters with health professionals*” [19] (p. 36). Whether, in general, “*in cases of alleged torture or ill-treatment, the best interests of the patient or alleged victim are often consistent with the purpose of the clinical evaluation, namely the effective documentation of torture and ill-treatment, which may corroborate an individual’s allegations of abuse*” [19] (p. 36), it is also true that in the case of migrant women, who are already in a fragile condition, the necessary—umpteenth—reenactment, for forensic purposes, of the ugliest, most difficult, humiliating moments of their lives, episodes that they would like to leave behind once and for all or bury in their memory, may be particularly difficult and more at risk of harming their condition. Moreover, the forensic/psychological assessment within this process cannot guarantee the confidentiality of the contents that emerged during the interviews with these women, as its purpose is precisely to certify the compatibility between what they say about the violence they have suffered and the diagnostic and forensic evidence.

To reduce and limit the risk of iatrogenic effects that these forensic operations may have on these women’s lives, it is necessary to reflect on the critical issues and contradictions that forensic doctors and psychologists must face in reconciling due respect for ethical principles and operational practice with the constraints (legal, administrative, bureaucratic, etc.) imposed by the asylum recognition process.

In this paper, we have chosen to focus on three dimensions, which, together with the ethical principles outlined above, must be carefully considered when assessing the psychophysical condition of migrant women victims of abuse and torture. These dimensions are “the time”, “the body”, and “the relationship”.

### 4.1. The Time Dimension: Between Kronos and Kairos 

In the experience conducted at our inward healthcare service, in line with international findings, the “time” dimension significantly influences the well-being of the asylum seeker during the process and may even affect the outcome of the application itself.

The timing of the bureaucratic asylum application process, in fact, hardly ever reflects the “internal time” of the applicant. Using a distinction made by the ancient Greeks, Kronos (the time of doing, of bureaucracy, etc.) and Kairos (the “internal” time, the “felt” time, the right or opportune moment) rarely coincide.

Sometimes, the timing of the asylum application process puts the applicant in the dangerous condition of having to relive, by repeatedly recounting—to doctors, lawyers, officials, the police, etc.—the violence suffered well before he/she is psychologically ready to go through it again, before he/she has had the chance to process the experiences connected to them, to mentalize them in a protected space (such as a psychotherapeutic space). This forcing of the subject’s “internal time”, if from a medico-legal point of view, may benefit the applicant because it allows an adequate assessment of the physical signs/outcomes of the torture and ill-treatment suffered; on the other hand, on the psychic level, it may unleash, exacerbate, or determine a condition of distress, even severe. In Italy, this circumstance may arise, in particular, in cases in which a “fast procedure” is established (the Art. 28 L.D. 25/2008, disciplines the Priority examination with reference to applications that are manifestly well-founded or that are submitted by vulnerable persons; the Art. 28-bis L.D. 25/2008, disciplines the accelerated procedure that concerns cases in which the application is presumed to have been submitted for dilatory or pretextual purposes) [36]. If the acceleration of procedure times exposes all migrants to the concrete risk of re-traumatization, this is even more true and serious for women, who are almost always victims of repeated violence, including sexual violence (often followed by terminated pregnancies), both in the country of origin and during the migration journey. In a short period of time, these women must repeatedly recall the terrible experiences they have undergone and, in the specific case of the medico-legal/psychological examination, even expose their bodies and their intimacy to strangers.

The time dimension affects significantly the principle of the applicant’s decision-making autonomy. Although it would be possible for the applicant to postpone the date of the medical/legal/psychological assessment visit (just as it would also be possible for justified reasons to postpone the hearing at the territorial commission) to an “opportune” moment, and even though this possibility is made known when the informed consent is signed, no woman has ever considered this possibility. There are numerous factors that may intervene in this choice, for example, a self-perception of “powerlessness” toward the complex bureaucratic machine, the fear of possible negative consequences on one’s evaluation process (denial), or the fear of having to wait too long.

Moreover, the risks connected to the “forcing of time” do not concern only the moment of the visit or the re-evocation but also the moment, probably much underestimated, of the restitution and the reading of the certification. In fact, the certification report, by putting the account of the violence suffered and the physical and psychic outcomes “on paper” makes them even more “concrete” and “real”. If the reading of the report takes place at the fulfilment of the Kairos, i.e., at the “right moment”, when the applicant is ready, this moment can prove to be of great importance as it gives the subject back possession of his own story, increases his self-agency, legitimizes his feeling and his suffering by formally and publicly acknowledging them. If, on the other hand, the reading of the relationship occurs when the subject is not ready when the time is not ripe, those fragile psychic defenses that allow him a precarious, shaky, partial cohesion of the self, and therefore a precarious psychic existence, will be undermined, configuring a condition of re-traumatization [37,38,39,40,41,42,43,44].

On the other side, the time dilation of the asylum procedure (to which the COVID-19 pandemic has also contributed) and the consequent feeling of “suspension” experienced by migrants can also have important negative consequences on the subjects’ mental health (anxiety, depression, etc.) [45,46,47,48,49,50,51,52,53,54,55], amplifying the psychological distress that is inherent in both the condition that Papadoupolos defines as “involuntary dislocation” [37] and the change of social role. These impact the outcomes of the medico-legal/psychological assessments and negatively orient the legal process. In general, asylum seekers are forced to live in reception centers while their applications for protection are processed. Numerous studies have shown that, in these facilities, although basic subsistence is guaranteed to them, the sense of security perceived by those accommodated becomes progressively lower as time passes, also due to several factors such as living in an uncomfortable/crowded/understaffed reception center; the absence of recreational and/or educational/qualifying activities; the reception center being positioned in an isolated location or poorly connected by public transport; frequent resettlement with a growing sense of disorientation and loneliness; and the loss of friendships or relationships with great difficulties established [51,56,57,58,59,60,61,62]. 

In addition, regarding the condition of women, the prolongation of the asylum application evaluation period contributes to increasing the risk of being subjected to verbal or physical violence or sexual harassment [63,64] and of being involved in prostitution-related crimes. For women forced to flee their country and who have left their children with relatives, the stress linked to the distance and the impossibility of supporting them without a regular income increases, as does the fear of not being able to start family reunification procedures. Finally, as regards pregnant women or those with babies (often born in Libya or shortly after they arrive in Italy), it is now known in the literature how to stay in a condition of continued uncertainty about one’s destiny, in a condition of psychophysical distress, and the sequelae of psychic traumas have a significantly negative impact on the mother-child relationship and the psychophysical development of children [65,66,67,68,69,70,71,72]. 

As mentioned above, the “prolongation” of the time of the procedure may affect the negative outcomes of the medico-legal/psychological assessment of asylum seekers. As far as the medico-legal assessment is concerned, as time passes, the physical signs of the violence potentially suffered are diminished or become less evident (as in the case of sexual assaults or rapes). Moreover, the subject may be assaulted after his/her arrival in the host country (e.g., in reception centers), and this makes it more difficult for forensic doctors to make an adequate assessment. As far as the psychological assessment is concerned, if the asylum seeker has been adequately psychotherapeutically supported, he/she may no longer show the condition of psychological distress that is presumed to be “typical” in traumatized subjects, with the consequent risk that his/her suffering will be underestimated by the Territorial Commission or by the appeal judge. If the asylum seeker has not been adequately supported, the condition of strong psychological distress, also connected to the above-mentioned factors, makes it extremely difficult to determine whether his/her suffering is attributable to the violence reported by the migrant concerning the migration route. 

### 4.2. The Body

In the context of the medico-legal/psychological assessment of the victims of ill-treatment and torture, “the body” takes on a particular significance as the subject-object of investigation, the memory of individual and human history, on which personal, cultural, religious, etc. issues insist. Often, it is the body that accounts for those emotions that the mind cannot manage because they are too overwhelming. The emotion denied by the mind thus finds a voice through the body, and the body speaks through tremendous headaches, chest pains, palpitations, sweating, muscle aches, a sense of alarm, an inability to relax, gastric disorders, acts of self-harm, etc. [42]. The activation of this “primitive system” [73] is functional at the level of self-preservation and survival of the individual, but since once triggered it is not easy to defuse, it becomes dysfunctional in the long term when the danger has ceased.

The women’s bodies, in particular, narrate the violence they have suffered, the wounds inflicted by the violence of the human act (on a material level), and the “symbolic” wounds inflicted by the policies, societies, and cultures of the countries of origin and reception. The exposure of the body, often full of scars, to a perceived inquisitive gaze creates a moment of extreme delicacy and criticality, especially in cases where the health team is not only composed of women. If the woman is not ready to expose herself or is not adequately prepared, the “inquisitive” and “indiscreet” gaze of the evaluator may evoke in her the experiences of extreme helplessness, degradation, and humiliation experienced during the violence/torture; a gaze to which it becomes extremely difficult to oppose a “no”, from which she must defend herself by keeping her jacket, scarf, and hat on—even if the room is well heated—as long as possible or until she feels she can trust it.

The exposure of a body that has suffered violence inevitably opens up issues of trauma and shame. Shame is one of the most complex intrapsychic processes as it touches fundamental dimensions of the self, identity, ego, and personality processes. In the migratory contexts referred to in this article, both trauma and shame are ubiquitous and inseparable, to the point that some scholars have proposed the concept of “post-traumatic shame” to emphasize the importance of this emotional condition and highlight its influence on the severity and course of PTSD symptoms [1,74,75,76,77,78,79]. Shame is at the core of the traumatic experience of sexual violence in its different forms: coercion, trafficking, genital mutilation, rape, slavery, and, finally, murder [80,81]. In FGM, the wound inflicted on the girls creates, in most cases, a corresponding psychic wound that will mark their life history and the perception of their own psycho-body continuity, health, sexuality, and subjectivity [12,81,82]. In our experience, some of these women have never carried out a gynecological examination precisely because of their difficulty in “making contact” with their wounded “physical and psychic body”. It is, therefore, even more difficult for them to agree to undergo a coroner’s examination for the assessment of FGM, even when this assessment could guarantee them the coveted international protection.

Rape, then, of which the majority of women from Sub-Saharan Africa are victims in Libya or during their migration route [83,84,85,86], is an extreme attack on the self that triggers unmanageable feelings of humiliation and shame [79,87]; it is the experience of the body being perceived as inherently damaged or contaminated and generating a sense of disconnection of the self from society. As several researchers point out, among people who have experienced trauma with bodily contact, especially sexual violence, shame plays a particular role in the development of peritraumatic or post-traumatic symptoms [79,88,89,90] and is linked to prolonged clinical problems [91,92].

Torture, physical and psychological violence, and mutilation constitute an attempt to annihilate women and a demonstration of power. Annihilation, in this case, does not coincide with killing but with depriving the woman of her dignity, erasing her identity, and dehumanizing her. The exposure of the self to the other through the body, in the context of the forensic/psychological examination, means for these women to symbolically “lay bare” the most intimate part of themselves and to narrate their shame resulting from violence and bodily abuse.

The writer Eduard Glissant [93] speaks of the right of the other “to opacity”, i.e., the right not to be totally understood and not to totally understand the other, as the protection of the individual being. According to Marozza [94], the concept of individuality is in the first place separateness, embankment, opacity, resistance to the gaze of others, the possibility of feeling and thinking differently from others, and the possibility of hiding and lying about one’s content. The awareness that no one can conquer one’s thoughts, affections, or feelings is just as reassuring, and perhaps even more fundamental, than the feeling that someone can understand them. 

However, the rigid “bureaucratization” of the pathway to recognition of the right to enjoy international protection does not contemplate the protection of the right to “opacity”, it does not contemplate confidentiality, and it does not take into account the pain and shame that one feels in recounting over and over again to perfect strangers the most horrible and barely thinkable episodes of one’s life and at feeling—moreover—the object of judgment.

### 4.3. The Relationship

The forensic/psychological examination is undoubtedly a moment of health care for the migrant subject, not only because—although it has a forensic connotation—the act is performed by specialized health personnel but also because it not infrequently represents the first moment when the asylum seeker undergoes a medical examination.

As also emphasized in the Istanbul Protocol [19], compared to medical doctors working in therapeutic contexts, forensic doctors and psychologists involved in certifying the outcomes of violence and torture have a different relationship with the persons they examine. Since they are generally held accountable for their observations and the outcome of their assessment becomes known to lawyers, judges, commissioners, etc., professional secrecy is not an integral part of their functions. In this context, therefore, the applicant’s power, autonomy, and freedom of choice are much more limited. Therefore, because of the delicacy of the assessment, the diminished autonomy of the asylum seeker, the marked asymmetry of power (real and perceived) that is inherent in the assessment process, and always in an attempt to minimize the consequences of possible re-traumatization, it is essential to reflect carefully on the manifest and underlying dimensions and variables that orient and determine the relational dynamic in the specifics of attesting to the outcomes of violence and torture.

In most cases, in an encounter between any health care team and a patient, the latter is willing to grant trust to the professionals he or she turns to. Trust is certainly central in contexts where patients’ health or even lives are at stake. For asylum seekers and women in particular, however, the issue is much more complex. For women migrants, trusting in health professionals during the forensic/psychological assessment process may prove extremely difficult, if not impossible. This is not only as a consequence of the multitude of potentially traumatic events they have been exposed to throughout their migration journey, but also because of the marked asymmetry of power that the assessment situation entails (and which can evoke the multiple forms of discrimination and subordination that these women very often experience in their countries) and the fears related to the outcome, the “judgement”. Moreover, the evaluation interview may induce fear and distrust as it may evoke previously experienced inquisitorial situations. 

As Theisen-Womersley [1] warns, “*being in the privileged and arguably hierarchically superior position of health professional in relation to refugee “patient” therefore comes with an ethical responsibility to be aware of the discrepancies in trauma narratives, and the power dynamics inherent to this context of which trust is a core component*” [1] (p. 281). To reduce the effects of possible re-traumatization, the healthcare team should be able to create a relational climate that is as relaxed, supportive, and non-judgmental as possible while being attentive to respecting the experiences and cultural background of asylum seekers; this will allow everyone present to experience the physical and relational space with less anxiety. Granieri [95] reminds us that to guarantee this form of reception, health workers must be able to “calibrate” at what distance to place themselves from the personal space—understood both in the physical and emotional sense—of the Other, since approaching it excessively or out of time, as mentioned above, can be experienced as an invasion, a violence.

Being in a relationship with the Other (from the psychological perspective, “Other” is understood as the “individual in his irreducible subjectivity”, bearer of his history, culture, personality, etc., and thus not superimposable or comparable to that of other individuals) does not simply entail an exchange of information between individuals (minds and bodies) present in the same room, but implies a series of “internal” modifications in all participants to the interaction that manifest themselves both at a bodily and a mental level, not always consciously. It is then necessary and ethically responsible, to reflect on the transferential and counter-transferential processes active in the encounter, on what defense mechanisms are put in place by the participants in the relationship, on the risk on the part of the health care workers of reification of the other, and on how the cultural dimension intervenes by orienting the interpersonal exchange and the psycho-body responses to trauma. Even the physical/institutional space in which the encounter takes place is not neutral; indeed, it intervenes significantly in determining the relational climate. The possibility of conducting the assessment interview at a hospital inward healthcare service, for example, as is the case in our experience, may allow the asylum seeker to recognize the health and not only forensic value of the assessment act; the hospital institution may also become in some way a guarantor of the role of the health professional and the correctness and non-influence by third parties of the assessment. Finally, the pre-interview phase of the presentation and signing of a detailed informed consent, which is compulsory in healthcare contexts, besides being a moment of clarification of doubts and a fundamental step in the construction of the relationship, restores to those who undergo the examination, even on a symbolic level, a part of that “decision-making power” which in the status recognition process is—in fact—denied to them, and contributes to creating a climate of trust.

#### 4.3.1. Transference, Counter-Transference, Defense Mechanisms and the Risk of “Reification” of the Other

As mentioned above, the interpersonal relationship implies a series of “internal” modifications in all participants of the interaction that manifest themselves both at a bodily and a mental level. In the particular context of severe trauma assessment, remembering, recounting, and listening to descriptions of violence and torture can produce intense emotional reactions in those who re-enact the events (such as distrust, fear, shame, anger, aggression, guilt, etc.) and in the health professionals who listen. 

The reciprocal emotional and bodily influence, mainly unconscious, typical of the transference-countertransference dynamic. The transference refers to often sudden actions and feelings whose origin can be traced back to past experiences, often linked to childhood, which, during the encounter, the asylum seeker manifests towards the health care team and which, therefore, seem to be directed towards the clinicians concerning the current situation; the counter-transference consists of the clinicians’ emotional and acted out response to the interviewee’s manifestations; the intensity and mode of the counter-transference response can also be traced back to previous experiences, in this case, of the health care professionals. If not adequately recognized and managed, they can compromise the assessment process. Certain counter-transferential reactions and the enactment of psychological defense mechanisms (such as identification, splitting, avoidance, denial, removal, projection, withdrawal, and indifference) in reaction to exposure to disturbing material may lead to over- or under-diagnosing the seriousness of the physical or psychological consequences of the violence suffered by the applicants. Fear of being deceived, in particular, may lead to an over-conservative approach to diagnosis; an over-identification or the fear of not being ‘adequate’ may, on the other hand, lead to an overestimation of the psychological sequelae of violence when the narrator’s pain is unconsciously amplified by that felt by the health care provider during the listening or by his belief that the experience of a severe trauma must necessarily lead to pathology even if unexpressed [18,19].

Feelings of anger or revulsion may occur not only towards the torturers but also towards the victim as a result of exposure to unusual and difficult-to-manage levels of anxiety and/or feelings of helplessness, which lead the healthcare professional “to unconsciously judge” the actions of asylum seekers. In the case of women, these counter-transference reactions may be evoked, for example, by accounts of rape, prostitution, or accounts describing the abandonment of their children [18,19,26], but also by their refusal to submit to checks that, as previously mentioned with regard to the GFM, could guarantee them international protection. The inability to recognize counter-transferential reactions on the part of the health professional may prevent the nature of certain particularly relevant patient manifestations from being grasped. It may be of great value for diagnostic purposes to understand, for example, whether the outbursts of anger are attributable to shame, guilt, fear, or vindication, whether the silence or a “hard” or “standoffish” attitude (these are frequent defense mechanisms among women who are victims of torture) is of a post-traumatic nature, an active choice not to interact with painful feelings [96,97] or, again, a manifestation of action and independence or rejection of gender or racial stereotypes [98].

A further risk that failure to reflect on counter-transferential and defensive responses entails is that of disregarding the subjectivity of the other that perturbs us (Freud, the Perturbing), reducing it to a “mere object” to be subjected to “judgement”. The dynamic underlying this defensive process, which devalues the humanity and uniqueness of the Other, evokes in the experience of the asylum seeker (and not only) the same mechanisms of dehumanization enacted by his persecutors and elicits, during the interview, responses marked by deep withdrawal and affective flattening or aggressive/oppositional reactions. 

The defensive “indifference” of health professionals risks making these victims victimized a second time (post-crime victimization) as a result of the unconditional coldness and disregard, aggressive attitudes, devaluation, incomprehension, and lack of protection with which they are treated not only in the context of the medical/legal psychological assessment but also by the police and members of the justice system, etc. Thus, the issue of “shame” returns to the forefront: for these women, the shame of feeling exposed (to the power of the other, in their fragility) and at the same time not being seen (by those who claim to want to help them), and the shame of being looked at (and feeling judged) where one would like to hide. 

Moreover, as Massari [99] points out, the dominant narratives on migration, trafficking in women, and prostitution often do not pay adequate attention to the beginnings of these stories, to the complexity of the intertwining of expectations and desires at the origin of migration projects, to the multiplicity of roles and social positions that the protagonists find themselves assuming, and to the forms of adaptation and resistance that they are often able to activate. Defining these women exclusively as slaves or victims runs the risk of obscuring and removing important subjective dimensions that make up the often extremely dramatic biographies of these women, who, despite being deeply marked and wounded by experiences of extreme abuse and exploitation, continue to claim the right to be considered subjects and not just objects of consumption and exchange. These are women who have made and continue to make choices, who do not want to conform to what has been decided for them, and who want to improve their living conditions. Even wanting to understand at all costs, wanting to explain, wanting to label the Other and her story, wanting to reduce the “distance”, even with the best of intentions, can mask an unconscious difficulty in really recognizing “the Other” in her “Otherness” and in her needs, which can have the paradoxical effect of making the Other more exposed and therefore more fragile. 

What has been said so far concerns only one aspect, certainly important and dramatic, of the lives of women arriving in Italy: the one that is most investigated and most known, and that risks wrongly defining them. All those who tend to come into contact with these women tend to absolutize the condition of victims, migrants, prostitutes, etc. and not to recognize them as “individuals”. The social representation of prostitutes or victims ends up engulfing any personal dimension.

#### 4.3.2. The Role of Cultural Differences in the Medico-Legal/Psychological Assessment

In the common imagination, when victims of violence and torture tell their dramatic stories, they must cry and despair and remember every horrendous detail; otherwise, they are inventing. The scientific literature on trauma [40,41,42,43,44] shows, however, that this is most often not the case: often some facts are not remembered but reconstructed; often the narration is monotonous, totally devoid of emotion, as if these people were recounting something trivial or that happened to others. 

This condition, “bizarre” at first glance, depends on the enactment of dissociative-type mechanisms, which entail a sort of emotional shutdown when the emotions experienced are so upsetting and overpowering that the mind cannot handle them without “burning out”. Dissociation is, therefore, essential for maintaining a minimum of “peace of mind” in those who experience dramatically extreme experiences. At the same time, dissociation is a dramatic and severe condition because it forces those who experience it to deny themselves the reality of their experience. In these cases, as mentioned earlier, the emotion denied by the mind finds a voice through the body, through tremendous headaches, chest pains, palpitations, sweating, muscle aches, a sense of alarm, an inability to relax, gastric disorders, acts of self-harm, etc. Individuals, therefore, may have trouble coherently recounting highly stressful or traumatic situations they have experienced.

Sometimes voluntary memories of events are not accessible to the victim, while involuntary and disturbing memories with flashbacks or nightmares occur spontaneously. But even when voluntary memories are retrieved, there may be circumstances or factors, such as shame or guilt, that may lead the individual not to provide detailed accounts [25]. Within this framework, cultural factors play an important role in the assessment of an asylum seeker, even more so in the case of women. Culture informs the emotional expression, norms, and outward manifestations of psychological distress, and numerous difficulties can arise in the assessment of an asylum seeker with a different cultural background from that of the health team [25,100,101,102,103,104]. 

One of the most relevant issues concerns the language barrier. In some cases, it may be difficult to find available cultural mediators; in other cases, the translation and interpretation process results in a loss of precision and nuance, with the result that inconsistent details and misunderstandings of meaning may appear in the narrative [26,105].

Many cultural factors relevant to asylum seeking that may emerge during assessments, if not acknowledged, may compromise the successful conclusion of the process itself. Though training in cultural competency can be useful for doctors and psychologists, it will always be extremely difficult for healthcare professionals to grasp the cultural specificities of subjects coming from such a diverse geographical area as the African one. For this reason, the presence of the cultural mediator is essential, with whom doctors and psychologists must establish a relationship of mutual trust, respect, and enhancement of professional skills. The cultural values attributed to types of violence, particularly violence against women, social interactions, and marital and child-rearing practices, can significantly influence descriptions of events. In the specific case of women, according to Elin Ekström, Ann-Christine Andersson, and Ulrika Börjesson [97], the European researchers debates on the lack of gender equality and violence perpetrated against women in non-Western cultures, undifferentiating diverse realities, especially about Islamic culture, paradoxically contributed to the victimization and subsequent stereotyping of Muslim migrant women. 

An emblematic example is represented by the FGM. The World Health Organization (WHO) defines the FGM “*all procedures that involve partial or total removal of the external female genitalia, or other injury to the female genital organs for non-medical reasons*. […] *The practice of FGM is recognized internationally as a violation of the human rights of girls and women. It reflects deep-rooted inequality between the sexes and constitutes an extreme form of discrimination against girls and women*” [106]. Not all women who have been subjected to this practice, however, share this definition, especially in cases where GFM is Type I, if it was performed when the women were infants, and in the (rare) cases in which the practice was performed by a doctor [107]. With reference to countries of origin, some studies show that, for women, FGM is seen not only as a marker of religious identity but also national identity, often marks a rite of passage, and ensures social status in the marital family and community [12,108,109]. Furthermore, in a context of exclusionary systems and experiences, such as socio-economic status or ethnicity at home or the migration and host country context, FGM can even represent a strong link to a personal gender, religious, and national identity and a way to cope with marginalization [110].

Therefore, the migrant woman, paradoxically, risks being deprived of the right to express her opinion and deprived of her free will as a partial or distorted understanding of her “culture” prevails over her point of view [97,111]. The cultural dimension also affects the narrative of symptoms and, indeed, the true experience of the physical, psychological/psychiatric disorder (in many cultures, for example, somatic expressions of psychiatric disorders predominate or include culture-specific expressions) [26,105]. 

Furthermore, psychic distress being “embedded” in a male or female body takes on different declinations depending on the sex of the individual and how this body is viewed within different cultures. 

These factors must be carefully weighed in the context of forensic evaluations, especially regarding issues of credibility and simulation. As also highlighted in the Istanbul Protocol [19] (p. 127), indeed, “*little published data exist on the use of projective and objective personality tests in the assessment of torture survivors and their use should therefore be evaluated with special care. There is no evidence that specific personality traits as measured in these tests typically result from the experience of torture or that certain personality traits are inconsistent with having been tortured. Also, psychological tests of personality lack cross-cultural validity. Personality tests have frequently been misused to stigmatize alleged victims, question their overall credibility or ascribe the emotional state to personality traits*”.

## 5. Conclusions

The importance of the medico-legal/psychological assessment of migrants who are victims of torture in their path to obtaining international protection is internationally recognized. Such an assessment can support the credibility of the asylum seeker through forensic evidence. At the same time, however, there is a risk that such forensic operations may have a negative impact on the lives of the most fragile subjects, such as women victims of torture. 

Based on the experience conducted at our inward healthcare service, this work has highlighted some critical issues that forensic doctors and psychologists have a duty to pay attention to during the assessment process. Especially in the case of women victims of torture, for example, it is extremely difficult to identify what is really “good for the applicant”, particularly when the risk of traumatization is extremely high and her decision-making autonomy is limited, as in the context of forensic assessment. Critical reflection on the difficulties and contradictions inherent in forensic evaluation relating to the need to reconcile due respect for the ethical principles (“beneficence”, “non-maleficence”, and “respect of autonomy”), the operating procedure, and the constraints imposed by the process of recognition of the right to asylum was carried out in three transversal dimensions. These three “lenses”, although not exhaustive for the complexity of the issue, are: time, body, and relationship. In our experience, they appear particularly relevant.

As highlighted in the work, it is essential that forensic doctors and psychologists carefully consider, in addition to purely technical issues, “external” factors (“time”, different cultural perspectives, etc.), knowledge, and “internal” factors for single individuals and the context (the dynamics of “power”, the transference-countertransference processes, expectations, stereotypes, and prejudices, etc.) guide and affect the interpersonal relationship, the assessment process and its outcome [19]. The absence of reflection on these critical issues, in addition to rendering the forensic evaluation invalid, constitutes a violation of ethical principles, a “reduction” of the Other from a subject and interlocutor to a mere “object of evaluation”. 

The generalization and homologation of the stories and lives of those who undergo a medical/psychological examination exponentially raises the risk of re-traumatization and re-victimization for all migrants, but even more for women. Unfortunately, there are many that we have heard in our inward healthcare service over the years. They are stories that only superficially appear similar, almost superimposable. In reality, if one is willing to listen more carefully, each of these stories is about women, about lives, and about choices on profoundly different paths. 

“Migrants” in general, and women specifically, are not a single, undifferentiated, and therefore invisible body but a multiplicity of people from different countries, from different socio-economic political contexts, with cultural and linguistic diversity. Their life experiences in the country of origin and/or during the migratory journey (as mentioned, sometimes superficially similar) mark the life of each individual in a unique, very personal way. Each assessment interview speaks to us profoundly about the irreducible subjectivity of the Other. 

Regarding women in particular, faced with the risk of absolutizing the condition of victims, migrants, prostitutes, of reducing the migrant experience within a single box, etc., it is possible, also thanks to a valid and relationally adequate medico-legal/psychological evaluation, to restore the wholeness of the migrant woman, to restore “thickness” to invisible bodies, to give value to suffering as an engine of transformation, and to welcome pain and life at the same time on the other hand, trying to account for the multiplicity of perspectives and experiences and their interconnections. As Gorman and Zakowski [112] (p. 44) point out “*the histories so variously told all revealed unique embodiments of suffering in the most personal and heartfelt of terms. More collective formulations risk blurring the distinctive contours of these individual accounts and obscuring the human face of each of the many survivors we have known. Instead, every report has been starkly singular; and in our judgment, it is not possible for anyone listening to such vital testimonies to reduce them simply to overarching themes and theories*”. 

In light of the foregoing, it therefore appears desirable that legal doctors, psychologists, and other health professionals potentially involved in the forensic evaluation of asylum seekers who are victims of torture are adequately trained. The training of clinical areas of expertise should not only cover specific professional practices in order to avoid technical errors [113] or implement a generic cultural competence. The training for forensic professionals should, above all, help them to recognize and face “emotionally” the difficulties and inconsistencies inherent in the assessment of the health status of asylum seekers, to recognize as much as possible how and how much their own evaluation is influenced by unconscious relational dynamics and subjective positions, and to exercise what Ponterotto [114] calls “vigilant ethical practice”, which we might consider a function of the forensic health professional’s self-awareness.

## Figures and Tables

**Figure 1 healthcare-11-02381-f001:**
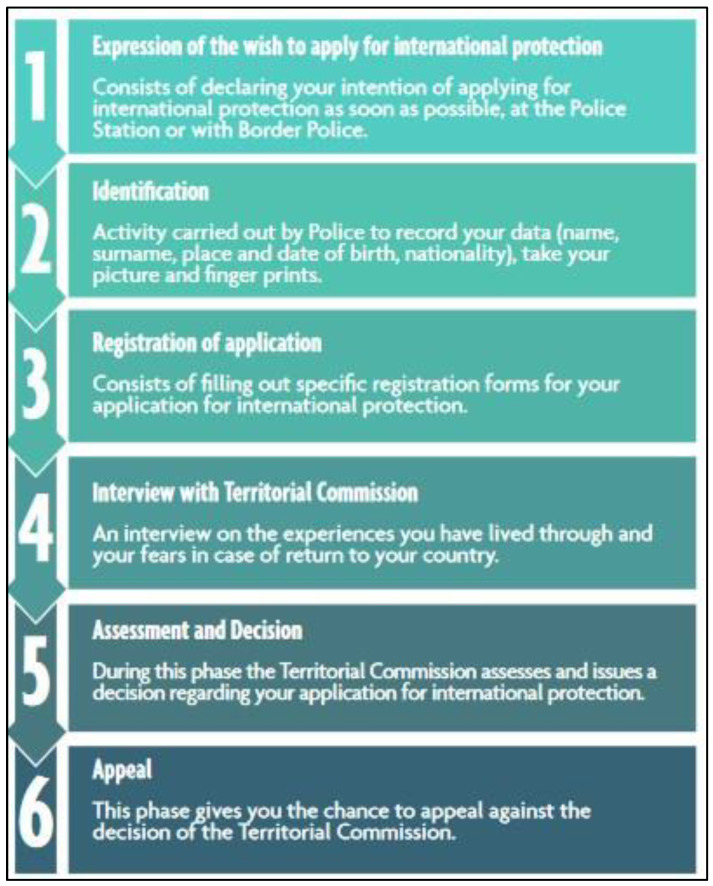
The process to be followed by asylum seekers [27].

## Data Availability

Data sharing is not applicable; no new data were created or analyzed.

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
