# Peer review of "Ethical and Forensic Issues in the Medico-Legal and Psychological Assessment of Women Asylum Seekers"

_healthcare, 2023, doi:10.3390/healthcare11172381_

Round 1

Reviewer 1 Report

The paper addresses important ethical and forensic issues in assessing vulnerable population of female asylum seekers. Experience-based reflections have been supported by current evidence and relevant literature in this issue. According to the article, professionals involved in such assessments should be adequately trained also in ethical issues related to their (forensic) evaluation. The final aim seems to be awareness of ethical aspects that may contribute to improved assessment quality.

Introduction includes references relevant for the paper. The aim of the paper has been clearly stated and motivated. Acronym (e.g. FGM) should be explained (what it stands for) when mentioned for the first time in the manuscript. Check also for other acronyms.

Section 2 presents some numbers about women accessing inward healthcare service. Is this publicly available/official statistics? According to the Informed consent statement there has been such consent statement for all subjects involved in the study. Also, ethical approval was not necessary for the study. The paper consists of ethical reflections and there is no individual data gathered for study population/obtained from individuals (qualitative/quantitative analysis was not the aim of the study) so it seems unclear why such consent has been necessary.

Ethical aspects of medical age assessment in the asylum process have been recently discussed in the literature (e. g. Erik Malmqvist Elisabeth FurbergLars Sandman Ethical aspects of medical age assessment in the asylum process: a Swedish perspective, Int J Legal Med 2018 May;132(3):815-823.).  Minors are included in female population that paper relates to but there are no such aspects (ethical issues of age assessment) mentioned. Is it because such assessments are not current for the described population or is it for some other reasons?

The training recommended for health professionals regards mainly ethics (lines 634-). Is there any training in cultural competency or other recommendations regarding section 4.3.2 that should be mentioned in the paper?

Text should be shortened. Repetitions of details may be avoided (e.g. symptoms described in line 512 and line 595). Summarize if possible or refer to previous section(s). Sentences are very long which effects readability.

Figure 1 – improve the quality of the figure if possible.

Author Response

Dear reviewer,

many thanks for your comments and suggestions. We have taken into account all of them in our revised paper. All changes made to the text and bibliography are in yellow. Here you have some more details.

Concerning:

  • Acronym (e.g. FGM) should be explained (what it stands for) when mentioned for the first time in the manuscript. Check also for other acronyms.

A: We have explained the Acronyms that are mentioned in the text.

  • Section 2 presents some numbers about women accessing inward healthcare service. Is this publicly available/official statistics? According to the Informed consent statement there has been such consent statement for all subjects involved in the study. Also, ethical approval was not necessary for the study. The paper consists of ethical reflections and there is no individual data gathered for study population/obtained from individuals (qualitative/quantitative analysis was not the aim of the study) so it seems unclear why such consent has been necessary.

A: The Italian legislation provides that each individual who undergoes a medical examination has to sign an informed consent. Although this paper consists of ethical reflections, informed consent and ethical approval have been requested by the editor. 

  • Ethical aspects of medical age assessment in the asylum process have been recently discussed in the literature (e. g. Malmqvist, E.; Furberg, E.; Sandman, L.  Ethical aspects of medical age assessment in the asylum process: a Swedish perspective. Int J Legal Med 2018, 132(3), 815-823.).  Minors are included in female population that paper relates to but there are no such aspects (ethical issues of age assessment) mentioned. Is it because such assessments are not current for the described population or is it for some other reasons?

A: Despite being a topic of great ethical importance, the authors have chosen not to address the issue of medical age assessment in the asylum process. This is for two reasons: 1) because in the Italian context, this procedure is not linked to the assessment of the results of violence and torture (the topic of the article); 2) because in Italy the legislation governing medical age assessment at the moment does not necessarily provide for the figure of the legal doctor and, for this reason, a great debate is underway. As regards, however, the assessment of the results of torture, the procedure used with minors is the same as that used with adults but, as mentioned in the text, the presence of the minor's legal guardian is mandatory.

  • The training recommended for health professionals regards mainly ethics (lines 634-). Is there any training in cultural competency or other recommendations regarding section 4.3.2 that should be mentioned in the paper?

A: Although training in cultural competency aimed at doctors and psychologists can be useful, it will always be extremely difficult for healthcare professionals to grasp the cultural specificities of subjects coming from such a diverse geographical area as the African one. For this reason, the cultural mediator is essential, with whom doctors and psychologists must establish a relationship of mutual trust, respect and enhancement of professional skills. This comment has been inserted into the text as a footnote.

  • Text should be shortened. Repetitions of details may be avoided (e.g. symptoms described in line 512 and line 595). Summarize if possible or refer to previous section(s). Sentences are very long which effects readability. We have rewritten many parts of the paper to make them shorter and easily to read.

A: Some sentences have been inserted as footnotes. Repetitions have been eliminated.

  • Figure 1 – improve the quality of the figure if possible.

A: The quality of the figure has been improved.

Reviewer 2 Report

The article is well written, coherent and will be an overall positive contribution to knowledge in this field. Some improvements are suggested for the article's soundness.

Formally, it is recommended to present the source of Figure 1 (page 3).

The article risks to be a bit descriptive, but the authors deepen some observation and further debate issues of time, body and relationship grounded in the ethics of healthcare with female asylum seekers. Some of the assumptions would gain with substantiation in other studies (even if extensive bibliography is reported in the majority of issues in the article, but not all), namely regarding the situation of women fleeing from extreme poverty (line 88), retaining that "the women's condition even more unbearable than that of men " - for which reasons, and what has been generally observed on this inequality in the movement of poor women and men? In line 328, it is stated "Rape then, to which the majority women from Sub-Saharan African are victims" - again, is there proof of this situation, or is this sentence derived from the medical experience of the authors (who report having attended less than 50 women along different years, a number possibly far inferior from the number of female asylum seekers arriving in Palermo in the same period)? Such generalizations risk to contribute to the stereotypes that the authors also try to combat in the article. The same reasoning applies to lines 453-3, "a "hard", "standoffish" attitude (frequent among women) ".

There is a tension that can be felt in diverse parts of the article regarding the importance of the individual in these medical assessments, which is correctly associated with the juridical fact that asylum seeking tends to be an individual procedure in which the person needs to prove what has happened in her/his specific life pathway. Yet, reinforcing the idea of "the Other", of an alterity process, even if reasonable from a psychological lens, can be a bit more challenged from recent readings of anthropology that would shed some new light on the debates that authors go through, also regarding cultural differentiation and the place of the self. All this, notwithstanding the good comprehension that asylum seekers (and other social groups) are perceived through broader social representations, namely by health professionals, as correctly identified by the authors. 

In lines 199 to 202, current scientific debates on the different experiences of time could be reported, namely from the literature on the mobilities turn and their impact on new ways to perceive and experience time in different social role situations.

Only minor corrections at the linguistic level are suggested, for example in line 274 "negatives outcomes", and in line 320 the lack of a point in the phrase transition "murder [76,77] In the FGM,"; so authors are advised to go through the whole text to proceed with such corrections.

Author Response

Dear reviewer,

many thanks for your comments and suggestions. We have taken into account all of them in our revised paper. All changes made to the text and bibliography are in yellow. Here you have some more details.

Concerning:

  • Formally, it is recommended to present the source of Figure 1 (page 3).

A: The bibliographic reference [25] had already been inserted in the course of the text next to the reference in Figure 1. For greater clarity, the bibliographic reference has also been added in the description of the figure.

  • Some of the assumptions would gain with substantiation in other studies (even if extensive bibliography is reported in the majority of issues in the article, but not all), namely regarding the situation of women fleeing from extreme poverty (line 88), retaining that "the women's condition even more unbearable than that of men " - for which reasons, and what has been generally observed on this inequality in the movement of poor women and men?

A: The phrase "the women's condition even more unbearable than that of men" refers to the greater difficulty of most African women in accessing finance or loans for starting work activities*, the profound gender disparities in work pay (when present), their impossibility (in some countries) to inherit and/or maintain possession of family assets (house, livestock, farmland etc.). These considerations have been added as a footnote to make the paper easy to read.

*(Techane, M.G. Economic Equality and Female Marginalisation in

the SDGs Era: Reflections on Economic Rights of Women in Africa. Peace

Human Rights Governance 2017, 1(3), 333-364. doi: 10.14658/pupj-phrg-2017-3-2; United Nations Development Programme: https://www.undp.org/?search=poverty+women+africa)

  • In line 328, it is stated "Rape then, to which the majority women from Sub-Saharan African are victims" - again, is there proof of this situation, or is this sentence derived from the medical experience of the authors (who report having attended less than 50 women along different years, a number possibly far inferior from the number of female asylum seekers arriving in Palermo in the same period)? Such generalizations risk to contribute to the stereotypes that the authors also try to combat in the article.

A: The statement reported on line 328 has now been better specified in the article. This statement derives from what was found in our inward healthcare service. It is in line with what is reported by the main international organizations that deal with the protection of human and migrant rights.** As regards the large number of women who access our service, this certainly does not include all the women asylum seekers present in Palermo, also because not all of them have been victims of torture. The ratio of approximately 1 woman for every 5 men (even if the trend is increasing) in the population of asylum seekers visited our clinic and, in any case, in line with the statistical data available on the Italian Government website https://www.interno.gov.it/it/stampa-e-comunicazione/dati-e-statistiche;

**(https://www.womensrefugeecommission.org/wp-content/uploads/2020/04/SVMB-Libya-Italy-ITALIAN-1.pdf; https://www.amnesty.it/libia-migranti-e-rifugiati-in-fuga-da-violenza-sessuale-persecuzione-e-sfruttamento/; https://www.unhcr.org/it/risorse/carta-di-roma/fact-checking/donne-rifugiate-la-violenza-molte-facce/; https://www.medicisenzafrontiere.it)

  • The same reasoning applies to lines 453-3, "a "hard", "standoffish" attitude (frequent among women) ".

A: Here the authors mean that "hard", and "standoffish" attitudes are a manifestation of defence mechanisms, known in the literature, referable to a condition of PTSD. This point has been better clarified in the text.

  • There is a tension that can be felt in diverse parts of the article regarding the importance of the individual in these medical assessments, which is correctly associated with the juridical fact that asylum seeking tends to be an individual procedure in which the person needs to prove what has happened in her/his specific life pathway. Yet, reinforcing the idea of "the Other", of an alterity process, even if reasonable from a psychological lens, can be a bit more challenged from recent readings of anthropology that would shed some new light on the debates that authors go through, also regarding cultural differentiation and the place of the self. All this, notwithstanding the good comprehension that asylum seekers (and other social groups) are perceived through broader social representations, namely by health professionals, as correctly identified by the authors. 

A: As well highlighted by the reviewer, the perspective used in this case is psychological. From the psychological perspective, "Other" is understood as the individual in his "irreducible subjectivity", bearer of his history, culture, personality, etc. and thus not superimposable or comparable to that of other individuals. This aspect has now been specified in the text in a footnote.

  • In lines 199 to 202, current scientific debates on the different experiences of time could be reported, namely from the literature on the mobilities turn and their impact on new ways to perceive and experience time in different social role situations.

A: A brief mention of the suggested theme has been introduced into the text in lines 243 to 246. It was decided not to further explore the topic because, although relevant, it is not central in the context of the medico-legal evaluation. We can insert the bibliographic references that the reviewer deems pertinent for the reader to deepen the topic.

  • Comments on the Quality of English Language: Only minor corrections at the linguistic level are suggested, for example in line 274 "negatives outcomes", and in line 320 the lack of a point in the phrase transition "murder [76,77] In the FGM,"; so authors are advised to go through the whole text to proceed with such corrections.

A: The text has been revised